# A Two-Sample Mendelian Randomization Analysis Investigates Associations Between Gut Microbiota and Celiac Disease

**DOI:** 10.3390/nu12051420

**Published:** 2020-05-14

**Authors:** Iraia García-Santisteban, Ariadna Cilleros-Portet, Elisabet Moyua-Ormazabal, Alexander Kurilshikov, Alexandra Zhernakova, Koldo Garcia-Etxebarria, Nora Fernandez-Jimenez, Jose Ramon Bilbao

**Affiliations:** 1Department of Genetics, Physical Anthropology and Animal Physiology, University of the Basque Country (UPV/EHU) and Biocruces-Bizkaia Health Research Institute, 48940 Leioa, Spain; iraia.garcia@ehu.eus (I.G.-S.); ariadna_cilleros001@ehu.eus (A.C.-P.); emoyua001@ikasle.ehu.eus (E.M.-O.); 2Department of Genetics, University of Groningen and University Medical Center Groningen, 9713 GZ Groningen, The Netherlands; alexa.kur@gmail.com (A.K.); sashazhernakova@gmail.com (A.Z.); 3Biodonostia, Gastrointestinal genetics group, 20014 San Sebastian, Spain; koldo.garcia@biodonostia.org; 4Spanish Biomedical Research Center in Liver and Digestive Diseases, CIBERehd, 28029 Madrid, Spain; 5Spanish Biomedical Research Center in Diabetes and associated Metabolic Disorders, CIBERDEM, 28029 Madrid, Spain

**Keywords:** celiac disease, gut microbiota, Mendelian randomization

## Abstract

Celiac disease (CeD) is a complex immune-mediated inflammatory condition triggered by the ingestion of gluten in genetically predisposed individuals. Literature suggests that alterations in gut microbiota composition and function precede the onset of CeD. Considering that microbiota is partly determined by host genetics, we speculated that the genetic makeup of CeD patients could elicit disease development through alterations in the intestinal microbiota. To evaluate potential causal relationships between gut microbiota and CeD, we performed a two-sample Mendelian randomization analysis (2SMR). Exposure data were obtained from the raw results of a previous genome-wide association study (GWAS) of gut microbiota and outcome data from summary statistics of CeD GWAS and Immunochip studies. We identified a number of putative associations between gut microbiota single nucleotide polymorphisms (SNPs) associated with CeD. Regarding bacterial composition, most of the associated SNPs were related to Firmicutes phylum, whose relative abundance has been previously reported to be altered in CeD patients. In terms of functional units, we linked a number of SNPs to several bacterial metabolic pathways that seemed to be related to CeD. Overall, this study represented the first 2SMR approach to elucidate the relationship between microbiome and CeD.

## 1. Introduction

Celiac disease (CeD), the most common food intolerance, is a chronic immune-mediated systemic disorder triggered by an aberrant immune response to dietary gluten in genetically predisposed individuals [1]. Virtually all CeD cases harbor specific genetic variants located in the human leukocyte antigen (HLA) region that encode for the HLA-DQ2/DQ8 heterodimers capable of presenting gluten peptides to T-cells, thus activating an inflammatory immune response in the intestine [2]. In addition to HLA risk alleles, genome-wide association studies (GWAS) have identified more than 40 non-HLA loci with modest contributions to CeD risk [3,4]. Whereas genetic predisposition and gluten exposure are necessary, they seem to be insufficient for the development of CeD, suggesting that other factors might serve as crucial triggers for disease onset and progression. Gut microbiota could be one such factor. 

The human gut microbiota is composed of the microbial communities, mainly bacteria, which live in the digestive tract. It is acquired at birth from the environment, and diversity builds up over the first few years of life [5,6]. Afterwards, the composition of the gut microbiota is largely shaped not only by environmental factors, such as diet [7], but also by host genetic components [8,9,10]. Studies carried out in twins support the idea that host genetics influences gut microbial diversity [11]. Specific genetic variations have been linked to a vast array of medical conditions [12], including CeD. Cross-sectional studies indicate that the composition of the gut microbiota is altered in subjects with CeD compared with controls [13,14,15,16]. Prospective studies performed so far report differences in microbiota composition and diversity between children with a genetic predisposition for CeD compared to those with a non-selected genetic background [17,18,19,20]. Considering that host genetics contributes to the composition and function of gut microbiota, it is feasible to think that risk genotypes for CeD act in part via the microbiota. However, the limited number of studies carried out in patients, together with the small sample sizes, make it difficult to mechanistically correlate host genetics, gut microbiota, and CeD. 

Mendelian randomization (MR) is an increasingly used statistical tool that can help establish a causal relationship between an exposure and an outcome of interest by employing single nucleotide polymorphisms (SNPs) as instrumental variables [21]. Two-sample Mendelian randomization (2SMR) refers to the application of MR in non-overlapping sets of individuals. These data can be obtained from public summary statistics of GWAS or estimated directly from own genomic datasets [22]. Landmark genomic studies have identified many SNPs associated with gut microbiota structure and function [8,9,10], which could serve as valid instruments for the exposure in 2SMR analyses. Regarding outcomes of interest, SNPs could be selected from relevant GWAS on CeD [4]. Thus, it is possible to investigate how gut microbiota composition and function may affect CeD risk by using publicly available data.

This study represented the first 2SMR approach to examine the interplay between host genetics, gut microbiota, and CeD. Our analysis identified a number of genetic variants associated with bacterial composition and function that could be driving CeD pathogenesis. These findings not only improve our understanding of the disease but also elicit new research lines towards diagnosis and clinical management. 

## 2. Materials and Methods 

Two-sample MR analysis was conducted using the “TwoSampleMR” R package [23], following the guidelines provided by the developers (https://mrcieu.github.io/TwoSampleMR), and in-house developed R scripts (available upon request). Figure 1a displays a flowchart describing the whole procedure. 

### 2.1. Preparation of Exposure Data

Genetic instruments from the exposure data were obtained from the raw results of the microbiome GWAS carried out by Bonder and colleagues in 1514 subjects [10]. Raw data comprised three independent datasets associating genetic variants (SNPs) with different bacterial traits, namely, taxonomies (taxa), MetaCyc pathways (pathway), and gene ontology (GO) terms. Bacterial taxonomies included information about phyla, classes, orders, families, genera, and species. MetaCyc pathways referred to experimentally elucidated metabolic pathways from bacteria retrieved from the MetaCyc metabolic pathway database (https://metacyc.org), while GO-terms comprised a subset of ‘informative’ GO terms, associated with more than 2000 proteins for which no descendant term was associated. A more detailed description is available in the original article from Bonder and collaborators [10]. We filtered the three SNP sets by *p*-value and further investigated only those showing suggestive associations using an empirical significance level (*p* < 10^−5^).

Next, datasets were converted to the “exposure data” format using the *format_data* function of the TwoSampleMR package. In this process, SNPs that did not have the standard “rs” nomenclature (less than 0.02% of all the variants) were dropped. In the case of SNPs that appeared more than once (due to association with more than one bacterial trait), only the one with the lowest *p*-value was retained.

Once exposure data had been formatted, independent SNPs were selected with the *clump_data* function in the same package, which used the 1000 genomes project as a reference panel. Stringent clumping criteria were set to retain only those SNPs with the lowest *p*-value above a linkage disequilibrium (LD) threshold of r^2^ = 0.01 in 10,000 Kb windows. 

### 2.2. Preparation of Outcome Data

Summary-level data of two independent association studies on CeD were retrieved from the IEU GWAS database. As a discovery dataset, we selected the largest-to-date GWAS on CeD, published by Dubois and colleagues in 2010 (ebi-a-GCST-000612). In this work, more than 500,000 SNPs were genotyped in 4533 CeD individuals and 10,750 controls [4]. As a replication dataset, we used the celiac Immunochip study, performed by Trynka and collaborators (ebi-a-GCST005523) in 2011. The investigators carried out a dense-genotyping of immune-mediated disease loci in 12,041 CeD subjects and 12,228 controls, mapping in-depth previously identified GWAS peaks and discovering several new CeD-associated regions [3]. Instrument SNPs from the GWAS and Immunochip results were obtained and formatted with the *extract_outcome_data* function, using previously prepared exposure data as input.

### 2.3. Harmonization of Exposure and Outcome Data

To ensure that the effect of an SNP on the exposure and on the outcome corresponds to the same allele, both exposure and outcome datasets were harmonized using the *harmonise_data* function. In this process, ambiguous and/or palindromic SNPs were removed, creating a new data frame that had the exposure and outcome data combined.

### 2.4. Two-Sample Mendelian Randomization (2SMR) and Statistical Analysis

2SMR analysis was performed on the harmonized data frame using the *mr()* function, which returned a data frame of estimates of the causal effect of the exposure on the outcome for a range of different MR methods (Wald ratio, MR Egger, weighted median, inverse variance weighted, and weighted mode). The multiple-testing-adjusted *p*-value, (false discovery rate, FDR) was calculated with the Benjamini–Hochberg procedure with the function *p.adjust()* of the “stats” package in R.

## 3. Results

Using a 2SMR approach, we set out to identify associations between gut microbiota (exposure) and CeD (outcome) using genetic variants. Exposure data were obtained from raw results of one of the most complete GWAS on the gut microbiome, where Bonder and colleagues examined the influence of host genetics on gut microbiota composition (by interrogating microbial taxonomies) and function (by assessing bacterial MetaCyc pathways and GO-terms) [10]. These data were used to prepare three exposure datasets: taxa, pathway, and GO, for our 2SMR analysis. After a stringent selection procedure of genetic variants (see Materials and Methods section for details), we retained 6756, 9179, and 9137 SNPs in taxa, pathway, and GO categories, respectively (Figure 1b). Outcome data was prepared using summary-level data from the to-date largest CeD GWAS, which included genetic variants encompassing the whole genome [4]. This led to the selection of 2232, 2865, and 2796 SNPs in the taxa, pathway, and GO categories, respectively, which were subsequently harmonized and finally analyzed with 2SMR. We selected SNPs with an FDR < 0.05, identifying 5, 6, and 1 hits associated with the taxa, pathway, and GO categories, respectively (Table 1).

To validate our results, we replicated the 2SMR analysis following the same criteria with the celiac Immunochip results [3] as outcome data (Appendix A). Since this study performed a dense genotyping of specific regions of the genome (including the HLA, the main CeD-predisposing locus), the number of SNPs finally analyzed, in this case, was around 10 times smaller, and the overlap with the SNPs included in Dubois analysis was low (Appendix A). Even with this small overlap, eight out of the nine significant SNPs from the discovery study were replicated in the analysis with the Immunochip data, and many of them showed suggestive associations at a genome-wide significance level (*p* < 10^−5^). Of note, the differences in *p*-values obtained in some of the hits were due to the different significance levels of those SNPs in the original CeD datasets used as outcome data. More importantly, forest plots comparing the beta values from both 2SMR analyses showed similar effect sizes and directions (Figure 2). Consistency in the results across the two independent 2SMR analyses built confidence in the obtained estimates.

The majority of the SNPs identified were located in introns and intergenic regions of CeD-related genes, and each of them was associated with one microbiome feature, either structural (taxa) or functional (pathway or GO) (Table 2). Traits related to bacterial composition were mainly linked to the Clostridiales order, a group of bacteria that have been observed to be altered in CeD individuals [24,25,26]. These data supported the link between the genetic variants and CeD through the modulation of gut microbiota diversity. Regarding functional units, associated bacterial pathways and GO-terms were related to the metabolism of certain amino acids or their derivatives, supporting the implication of microbial metabolism on CeD pathogenesis. 

## 4. Discussion

To our knowledge, this work represented the first attempt to explore the interplay between host genetics, gut microbiota, and CeD using a 2SMR approach. One of the key points for performing a successful 2SMR analysis is the appropriate selection of datasets to be used as exposure and outcome. In our study, instead of limiting the exposure data to selected instruments, we included the complete summary statistics of the microbiome GWAS by Bonder and collaborators in our analysis, thus increasing the coverage of our study. We then applied a stringent clumping protocol that selected genome-wide significant and independent variants in 10,000 Kb windows, to end up with less than 3% of the genomic variants from the original datasets. In addition, compared to other genome-wide studies on gut microbiome [8,9], the GWAS from Bonder and colleagues evaluates the contribution of host genetics not only to bacterial composition but also in terms of functional units [10]. In fact, recent whole-metagenome shotgun sequencing has revealed that fecal metabolic profiles are associated with only a few key species but with many common microbial functions, stressing the importance of the functional role of microbial communities on top of the microbial species present in the flora [27]. 

Regarding outcome data, we selected two breakthrough studies on CeD: the GWAS performed by Dubois and collaborators was selected as a discovery dataset since it covered genetic variants encompassing the whole genome [4]. As a “replication” dataset, we selected the celiac Immunochip study that contains a smaller number of SNPs from regions known to be associated with immune disorders but performs a much denser genotyping of those particular loci [3]. Despite the limited overlap between the SNPs included in each of the two 2SMR analyses, eight out of the nine significant associations from the discovery study were replicated, with similar effect sizes, underscoring the validity of our results. On the other hand, the SNPs identified were located mainly in immune-related loci, and this reinforced the idea that the host immune system plays a relevant role in shaping microbial communities [28] also in the context of CeD.

In our analysis, we pinpointed a number of significant associations between host genetics, gut microbiome, and CeD. One of the most interesting findings was that all the identified taxa-related SNPs were linked to the Firmicutes and Proteobacteria phyla, two groups of bacteria whose abundance has been shown to be altered in CeD patients that harbor HLA risk alleles [17,24,29,30,31]. In this sense, it has been shown that Type I diabetes risk variants in the IL2 pathway genes are associated with microbial shifts in mice and humans, including the decrease of the *Lachnospiraceae* and *Clostridiales* families of the Firmicutes phylum [32]. Additionally, the oral administration of these particular strains is able to reduce disease severity in mouse models of colitis and allergic diarrhea [33], possibly through a bacteria-induced upregulation of *ICOS* and a consequent increased production of regulatory T-cells that reduce exacerbated immune responses and inflammation. In the present work, we identified rs7594065 and rs6848139 (located close to the well-known interleukin-2 pathway genes—*ICOS*/*CTLA4* and *IL-2—*respectively) to be associated with both microbial features and CeD. In particular, the CeD-risk allele of rs7594065 appeared to be negatively correlated with *Clostridiales* abundance. We proposed that these SNPs somehow reduce the presence of *ICOS* and regulatory T cell-inducing strains in the microbial flora, contributing to the characteristic inflammation of the celiac intestine. 

Another SNP from the pathway category, namely rs131659, is associated with the bacterial arginine/polyamine biosynthesis pathway. As illustrated in Figure 3a, the first step of this pathway is the conversion of arginine to ornithine by arginase, an enzyme whose activity has been shown to be induced by gluten peptides in human monocytes. In the second step, L-ornithine carboxy-lyase (a GO-term that had also been identified in our analysis) transforms ornithine into polyamines, which have been reported to increase permeability and inflammation *in vitro* [34]. Gluten peptides exert the same level of activation of the arginine metabolism in CeD and healthy individuals [35], suggesting that the increased activity of this pathway corresponds to the celiac microbiota and, in turn, causes increased permeability and inflammation of the intestine.

The last example among our functional candidates is rs11867190, which is associated with increased L-lysine biosynthesis (Figure 3b). A study carried out with gluten treated with transglutaminase type 2 (TG2) in the presence of a saturating amount of L-lysine has shown an interruption of the deamidation reaction catalyzed by TG2 and, consequently, a lower affinity of these gluten-derived peptides for HLA-DQ2 molecules. In fact, enzymatic modification of gluten with TG2 plus lysine has been able to suppress its immunologic effects on the duodenal mucosa of CeD patients [36]. These data suggest that a lysine-producing flora in CeD patients could represent a mechanism by which TG2 activity is weakened as a response to the disease. 

Nevertheless, some limitations of this study should also be considered. Our study used microbiome data from a cohort of adult individuals, whereas several microbiome studies on CeD have been carried out in children. It would be interesting to perform a similar analysis in an infant population when genome-wide studies on childhood microbiota become available. Also, we cannot exclude the possibility that non-significant associations might arise from a lack of power due to the limited sample size (1514 individuals) of the microbiome GWAS. In this sense, the recently established MiBioGen consortium aims to meta-analyze large-scale data from 18 independent cohorts in order to investigate host genetics and microbiota associations in almost 20,000 subjects. This large-scale resource will also be very useful to assess the biological impact of gene-microbiome interactions in different diseases, including CeD [37]. Finally, the 2SMR approach cannot rule out the possibility that the associations discovered are pleiotropic rather than causal, and even then, one cannot fully discard the possibility of reverse causation. Actually, we proposed two opposite models: one in which variants associate with CeD through the modification of bacterial arginine and polyamine synthesis and/or through the reduction of regulatory T cell-inducing bacterial strains (direct causation), as well as another in which SNPs are associated with CeD and the bacterial flora adapts to the pathological condition of the host by producing lysine in an attempt to impair gliadin presentation by HLA molecules (reverse causation). Further studies will certainly be needed in order to clarify this complex scenario.

In summary, this was the first work to identify genetic variants that could mediate CeD pathogenesis through gut microbiota. Our results should be interpreted with caution, pending epidemiological and experimental confirmation of the mechanisms proposed, but put forward interesting and plausible hypotheses that can explain the complex interactions between the host and microbial communities in the celiac gut, which might be potentially useful for the prediction, prevention, and treatment of the disease.

## Figures and Tables

**Figure 1 nutrients-12-01420-f001:**
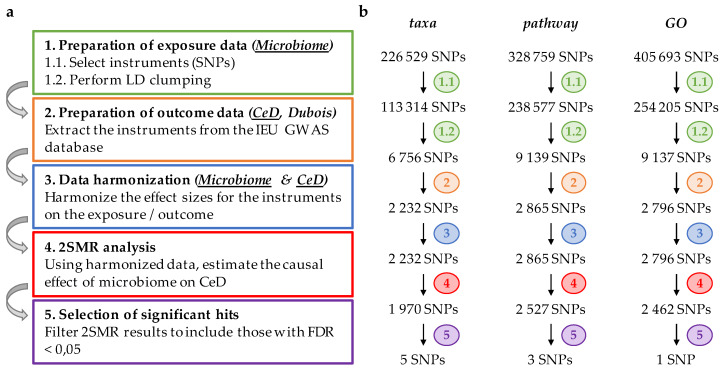
Schematic representation of two-sample Mendelian randomization analysis (2SMR) using Bonder microbiome and Dubois celiac disease (CeD) genome-wide association study (GWAS) as exposure and outcome datasets, respectively. (**a**) Flowchart of the step-by-step analysis: after preparing exposure data (Step 1), outcome data is extracted (Step 2), both datasets are harmonized (Step 3), and 2SMR analysis is performed (Step 4); finally, significant hits are selected based on their false discovery rate (FDR) (Step 5); (**b**) Diagram, representing the number of single nucleotide polymorphisms (SNPs) selected in each category (taxa, pathway, gene ontology (GO)) after performing each step of the analysis.

**Figure 2 nutrients-12-01420-f002:**
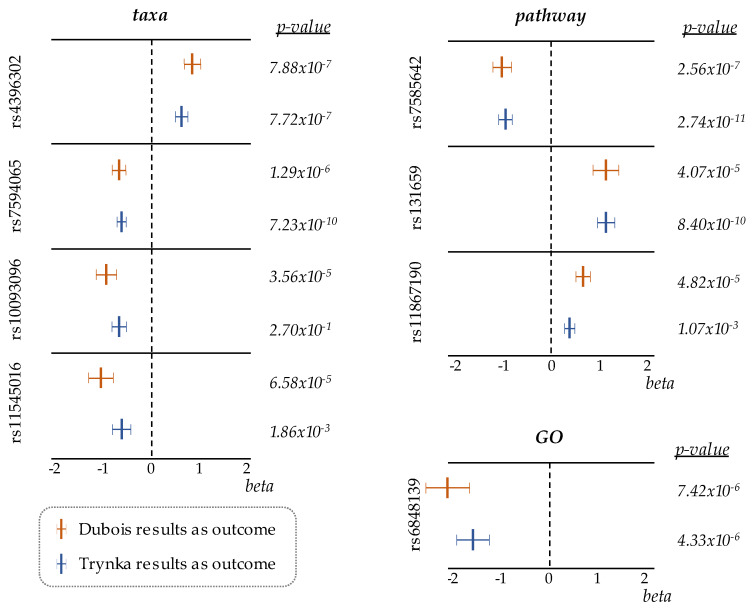
Forest plots, comparing effect-sizes and *p*-values of SNPs identified in taxa, pathway, and GO categories, in two independent 2SMR analyses where either Dubois or Trynka CeD datasets were used as outcome data.

**Figure 3 nutrients-12-01420-f003:**
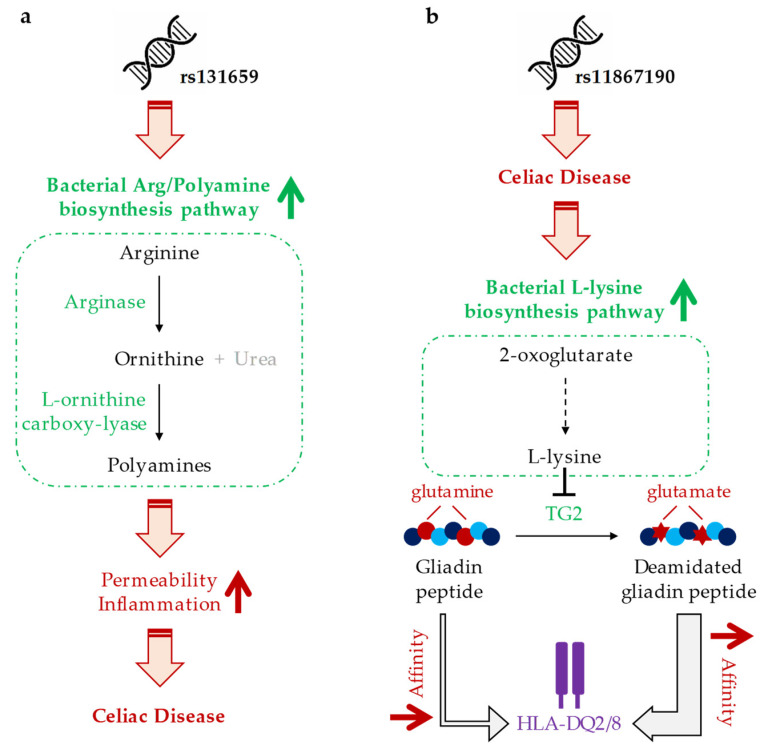
Schematic representation of the hypothetical mechanism of action of the identified SNP-microbiota associations. (**a**) rs131659 is associated with increased bacterial arginine/polyamine biosynthesis pathway, where arginine is converted to polyamines; by increasing permeability and inflammation, polyamines could play a role in CeD manifestation. (**b**) rs11867190 SNP is associated with CeD; increased production of L-lysine amino acid in the celiac intestine could be an adaptation of gut bacteria to counteract the activity of type 2 transglutaminase (TG2), which converts glutamines into glutamates; impaired TG2 activity would prevent the increase in the affinity of gliadin peptides for HLA-DQ2/8 receptors. HLA, human leukocyte antigen.

**Table 1 nutrients-12-01420-t001:** Two-sample Mendelian randomization estimates between the gut microbiota and CeD.

SNP	Effect/Other Alleles	Chr.	Position	*p*-Value	Effect size ± SE	FDR
**taxa**						
rs4396302	A/G	11	128420926	7.88 × 10^−7^	0.80 ± 0.16	0.001
rs7594065	T/C	2	204814676	1.29 × 10^−6^	−0.61 ± 0.13	0.001
rs10093096	C/T	8	64907701	3.56 × 10^−5^	−0.84 ± 0.20	0.027
rs11545016	T/C	8	22438313	6.58 × 10^−5^	−0.96 ± 0.24	0.037
rs12913063	T/C	15	75424593	9.97 × 10^−5^	1.09 ± 0.28	0.044
**pathway**						
rs7585642	A/C	2	61217542	2.56 × 10^−7^	−0.92 ± 0.18	0.001
rs131659	G/A	22	21964761	4.07 × 10^−5^	1.04 ± 0.25	0.046
rs11867190	A/G	17	5261220	4.82 × 10^−5^	0.59 ± 0.14	0.046
**GO**						
rs6848139	C/A	4	123395041	7.42 × 10^−6^	−1.95 ± 0.43	0.021

Chr.: Chromosome; SE: Standard error; FDR: False discovery rate.

**Table 2 nutrients-12-01420-t002:** Associated microbiota traits of 2SMR hits.

SNP	Associated Microbiota Trait
**taxa**	
rs4396302	Firmicutes (*p*), Clostridia (*c*), Clostridiales (*o*), *Peptostreptococcaceae* (*f*), *Peptostreptococcaceae* (*g*), *Peptostreptococcaceae* unclassified (*s*)
rs7594065	Firmicutes (*p*), Clostridia (*c*), Clostridiales (*o*), *Clostridiales* noname (*f*), *Pseudoflavonifractor* (*g*)
rs10093096	Proteobacteria (*p*)
rs11545016	Firmicutes (*p*), Clostridia (*c*), Clostridiales (*o*), *Lachnospiraceae* (*f*), *Lachnospiraceae* noname (*g*)
**pathway**	
rs7585642	PWY-6060 (malonate degradation II, biotin-dependent)
rs131659	ARG+POLYAMINE-SYN (super pathway of arginine and polyamine biosynthesis)
rs11867190	PWY-3081 (*L*-lysine biosynthesis *V*)
**GO**	
rs6848139	GO:0016831 (MF, carboxy-lyase activity)

*p*: phylum; *c*: class; *o*: order; *f*: family; *g*: genus; *s*: species.

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
