# Peer review of "A Two-Sample Mendelian Randomization Analysis Investigates Associations Between Gut Microbiota and Celiac Disease"

_nutrients, 2020, doi:10.3390/nu12051420_

Round 1

Reviewer 1 Report

In the manuscript “A two-sample Mendelian Randomization analysis investigates associations between gut microbiota and celiac disease,” the authors used a Genome wide association study (GWAS) results as exposure data and two studies, one GWAS study and one Immunochip study, as outcome data. In their analysis, they linked a number of SNPs to several bacterial metabolic pathways that seem to be related to celiac disease.

 The major concern is: why the authors chose p<10-5 as cutoff value for significance rather than the P < 5.0 x10-8, the common genome-wide significance cutoff? Some discrepancies among the p values were noted between the dataset from Dubois study and Trynka. For example, the p values for rs131659 in the pathway association were 4.07x10-5 and 8.40x10-10. The authors should provide explanation for this and proved rationale that why the significance analysis is valid.

Reviewer 2 Report

The manuscript „A Two-Sample Mendelian Randomization analysis investigates associations between gut microbiota and celiac disease” is generally well-written and the topic seems to be interesting for the readers. The obtained results have been deeply discussed. The limitations of the study have been indicated by the Authors in the Discussion section.

Reviewer 3 Report

I’ve read with attention the paper of García-Santisteban et al. that is potentially of interest. The background and aim of the study have been clearly defined. The methodology applied is overall correct, the results are reliable and adequately discussed. The figures are well drawn.  As a really minor comment, I've found a typo in table 2.

Round 2

Reviewer 1 Report

The authors have addressed my questions.